# Sequential immunotherapy and targeted therapy for metastatic BRAF V600 mutated melanoma: 4-year survival and biomarkers evaluation from the phase II SECOMBIT trial

No prospective data were available prior to 2021 to inform selection between combination BRAF and MEK inhibition versus dual blockade of programmed cell death protein-1 (PD-1) and cytotoxic T lymphocyte antigen-4 (CTLA-4) as first-line treatment options for *BRAF*V600-mutant melanoma. SECOMBIT (NCT02631447) was a randomized, three-arm, noncomparative phase II trial in which patients were randomized to one of two sequences with immunotherapy or targeted therapy first, with a third arm in which an 8-week induction course of targeted therapy followed by a planned switch to immunotherapy was the first treatment. BRAF/MEK inhibitors were encorafenib plus binimetinib and checkpoint inhibitors ipilimumab plus nivolumab. Primary outcome of overall survival was previously reported, demonstrating improved survival with immunotherapy administered until progression and followed by BRAF/MEK inhibition. Here we report 4-year survival outcomes, confirming long-term benefit with first-line immunotherapy. We also describe preliminary results of predefined biomarkers analyses that identify a trend toward improved 4-year overall survival and total progression-free survival in patients with loss-of-function mutations affecting *JAK* or low baseline levels of serum interferon gamma (IFNγ). These long-term survival outcomes confirm immunotherapy as the preferred first-line treatment approach for most patients with *BRAF*V600-mutant metastatic melanoma, and the biomarker analyses are hypothesis-generating for future investigations of predictors of durable benefit with dual checkpoint blockade and targeted therapy.

Combination BRAF and MEK inhibition[1,2] as well as dual blockade of programmed cell death protein-1 (PD-1) and cytotoxic T lymphocyte antigen-4 (CTLA-4)[3,4] both offer clinical benefit and are approved by the United States (US) Food and Drug Administration (FDA) and the European Medical Agency (EMA) as first-line treatment options for *BRA*FV600-mutant melanoma. Until 2021, no prospective data were available to inform selection between first-line targeted therapy versus immunotherapy for the 40–50% of patients with cutaneous melanoma

with *BRAF*V600-mutant tumors[5], though retrospective analyses[5–7] and preclinical data[8–12] indicated that checkpoint blockade should be offered prior to BRAF/MEK inhibition. The randomized trials SECOMBIT, which included an arm investigating a planned switch to immunotherapy after an 8-week induction course of targeted therapy[13], and DREAMseq[14] established immunotherapy as the preferred first-line treatment approach, demonstrating improved response rates to immunotherapy and prolonged survival in patients who received the

✉e-mail: p.ascierto@istitutotumori.na.it

anti-CTLA-4 ipilimumab plus the anti-PD-1 nivolumab until progressive disease (PD) and subsequent BRAF/MEK inhibition compared to those treated with the reverse sequence.

Clinical benefit with immunotherapy is known to be delayed, however, and some patients with aggressive and broadly disseminated disease may not have efficiently functioning immune systems or enough time to wait for immune-mediated tumor clearance. SECOMBIT included a "sandwich" arm in which patients received 8 weeks of the BRAF inhibitor encorafenib with the MEK inhibitor binimetinib before a planned switch to ipilimumab plus nivolumab, yet the optimal criteria to select patients for the sandwich approach are still not known. Potential biomarkers include elevated serum lactate dehydrogenase (LDH), which is indicative of a glucose-starved and hypoxic tumor microenvironment (TME)[15–18], and defective interferon gamma (IFNγ) signaling due to loss-of-function mutations affecting *JAK*, which is considered a major mechanism of resistance to anti-PD-1[19,20], however, the cytokine is highly pleiotropic and may cause immunosuppression via a number of mechanisms including inhibition of natural killer and CD8$^+$ T cell effector functions, deletion of tumor antigen-specific T cells, and induction of tolerogenic dendritic cells[21–26]. Here, we report 4-year outcomes from the randomized, open-label, phase II SECOMBIT trial (NCT02631447), as well as preliminary biomarkers analyses indicating trends toward improved survival in the immunotherapy-first and sandwich arms among patients with elevated LDH, low serum IFNγ, and deleterious mutations in *JAK*.

## Results

### Patients and treatment

Between November 2016 and May 2019, 251 patients with untreated, metastatic *BRAFV600*-mutant melanoma were screened. A total of 209 patients from 37 sites in 9 countries were enrolled and randomized across the three treatment arms: 69 in Arm A (encorafenib plus binimetinib until PD followed by ipilimumab plus nivolumab), 71 in Arm B (ipilimumab plus nivolumab until PD followed by encorafenib plus binimetinib), and 69 in Arm C ("sandwich," encorafenib plus binimetinib for 8 weeks followed by ipilimumab plus nivolumab until PD followed by encorafenib plus binimetinib). The median age of patients in Arms A, B, and C was 55.0 (range 19–77), 55.0 (range 18–81), and 51.0 (range 28–80), with 60.9%, 47.9% and 60.9% of male sex, respectively. Tumor stage was not known for one patient in arm B and one in arm C (Table 1).

As of June, 2022, 4 years from treatment, among the 206 patients who received at least 1 dose of the study sequence ($n = 69$, $n = 69$, and $n = 68$ in Arms A, B, and C, respectively), 64 remained on treatment ($n = 17$, $n = 24$, and $n = 23$ in Arms A, B, and C, respectively). The numbers of patients who completed the entire sequence (ie, PD on treatment 1 and on treatment 2) were 19, 10, and 20 in Arms A, B, and C, respectively. During treatment across the arms, there were 13 deaths in Arm A (7 during encorafenib plus binimetinib treatment and 6 during ipilimumab plus nivolumab treatment), 11 deaths in Arm B (3 during the ipilimumab plus nivolumab treatment and 8 during encorafenib plus binimetinib treatment), and 4 deaths in Arm C (2 during the ipilimumab plus nivolumab treatment and 2 during second treatment of encorafenib plus binimetinib treatment). Adverse events led to treatment discontinuation in 11 patients in Arm A, 10 patients in Arm B, and 11 patients in Arm C (Supplementary Fig. 1).

### 4-year survival outcomes

The primary analysis was reported previously (5). With an additional 13 months of follow-up (median 43 months, IQR: 37–51), the 3-year total progression free survival (TPFS, time from randomization until second progression) rates for Arms A, B, and C were 34% (95% CI 24–46), 55% (95% CI 43–67), and 54% (95% CI 42–66), respectively. TPFS rates at 4 years were 29% (95% CI 18–40), 55% (95% CI 43–67), and 54% (95% CI 42–66) for Arms A, B, and C (Supplementary Table 1, Fig. 1A).

OS rates at 3 and 4 years, respectively, were 53% (95% CI 41–65) and 46% (95% CI 33–59) for Arm A, 64% (95% CI 53–76) and 64% (95% CI 53–76) for Arm B, and 61% (95% CI 50–73) and 59% (95% CI 47–71) for Arm C (Fig. 1B). Although The SECOMBIT trial was not designed as a comparative trial and no P value calculation was planned, P values for TPFS and OS between the arms are reported in Supplementary Table 1.

### Interaction between adverse prognostic features and 4-year survival

Across Arms A, B, and C, 43 (62.3%), 41 (57.7%) and 43 (62.3%), patients had <3 metastatic sites, respectively. The numbers of patients with ≥3 metastatic sites were 25 (36.2%), 29 (40.9%), and 25 (36.2%) in Arms, A, B, and C. Brain metastases were present in 2 patients, 1 in Arm B and 1 in Arm C. LDH was ≤1 × ULN in 41 (59.4%), 37 (52.1%), and 44 (63.8%) of patients in Arms A, B, and C, respectively. In total, 28 (40.6%), 34 (47.9%), and 25 (36.2%) of patients in Arms A, B, and C, respectively, had LDH > 1 × ULN. Among the patients with elevated LDH, the levels were >2 × ULN in 7 (10.1%), 9 (12.7%), and 7 (10.1%) patients across arms A, B, and C (Table 1).

The 4-year TPFS rates for patients with <3 metastatic sites were 33% (95% CI 19–48) in Arm A, 59% (95% CI 43–74) in Arm B, and 59% (95% CI 44–74) in Arm C (Supplementary Fig. 2). Patients with ≥3 metastatic sites had 4-year TPFS rates of 23% (95% CI 7–39), 51% (95% CI 32–69), and 46% (95% CI 27–65) in Arms A, B, and C, respectively. OS rates at 4 years were 55% (95% CI 39–72) vs 32% (95% CI 12–51), 65% (95% CI 50–81) vs 63% (95% CI 44–81), and 62% (95% CI 47–78) vs 54% (95% CI 34–73) for patients with <3 compared to ≥3 metastatic sites in Arms A, B, and C, respectively (Supplementary Fig. 2).

The 4-year TPSF rates for patients with elevated versus normal LDH were 18% (95% CI 3–34) vs 31% (95% CI 14–48) in Arm A, 48% (95% CI 29–67) vs 58% (95% CI 42–73) in Arm B, and 60% (95% CI 38–82) vs 51% (95% CI 36–65) in Arm C. OS followed a similar pattern, with OS rates at 4 years for patients with elevated compared to normal LDH of 42% (95% CI 22–61) vs 53% (95% CI 37–70) in Arm A, 53% (95% CI 33–73) vs 70% (95% CI 55–85) in Arm B, and 65% (95% CI 44–86) vs 56% (95% CI 41–70) in Arm C (Supplementary Fig. 3).

### Tumor and peripheral biomarkers analyses

NGS was performed on tumor tissue obtained at baseline from 83 patients, with 29 from Arm A, 25 from Arm B, and 30 from Arm C included in the analysis for TMB. Analysis of deleterious *JAK* mutations

## Table 1 | Baseline characteristics of the intention-to-treat population

| | Arm A (*n* = 69) | Arm B (*n* = 71) | Arm C (*n* = 69) |
|---|---|---|---|
| Median age, years (range) | 55.0 (19–77) | 55.0 (18–81) | 51.0 (28–80) |
| Gender – Male, *n* (%) | 42 (60.9%) | 34 (47.9%) | 42 (60.9%) |
| ECOG-PS 0, *n* (%) | 57 (82.6%) | 62 (87.3%) | 62 (89.9%) |
| Lactate Dehydrogenase (LDH) levels, *n* (%) | | | |
| ≤1.00 × ULN | 41 (59.4%) | 37 (52.1%) | 44 (63.8%) |
| >1.00 × ULN | 28 (40.6%) | 34 (47.9%) | 25 (36.2%) |
| >2.00 × ULN | 7 (10.1%) | 9 (12.7%) | 7 (10.1%) |
| Stage, *n* (%) | | | |
| M0-M1a–M1b | 29 (42%) | 28 (39.4 %) | 29 (42%) |
| M1c | 40 (58.0%) | 42 (59.1%) | 39 (56.5%) |
| Not reported | 0 | 1 (1.5 %) | 1 (1.5%) |
| Number of metastatic sites, *n* (%) | | | |
| <3 | 43 (62.3%) | 41 (57.7%) | 43 (62.3%) |
| ≥3 | 25 (36.2%) | 29 (40.9%) | 25 (36.2%) |
| Not evaluated | 1 (1.5%) | 1 (1.4%) | 1 (1.5%) |

Stage is reported as described in the American Joint Commission on Cancer Cancer Staging Manual, Version 7.

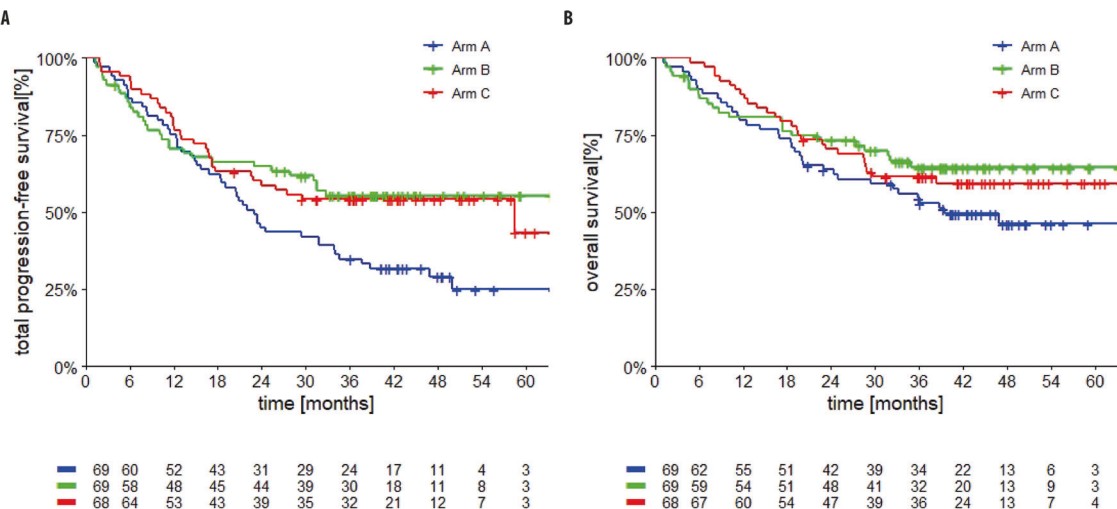

**Fig. 1 | Kaplan–Meier survival curves with 3-year and 4-year rates for Arm A (blue), Arm B (green), and Arm C (red). A** Total progression free survival; **B** Overall survival. Source data are provided as a Source Data file.

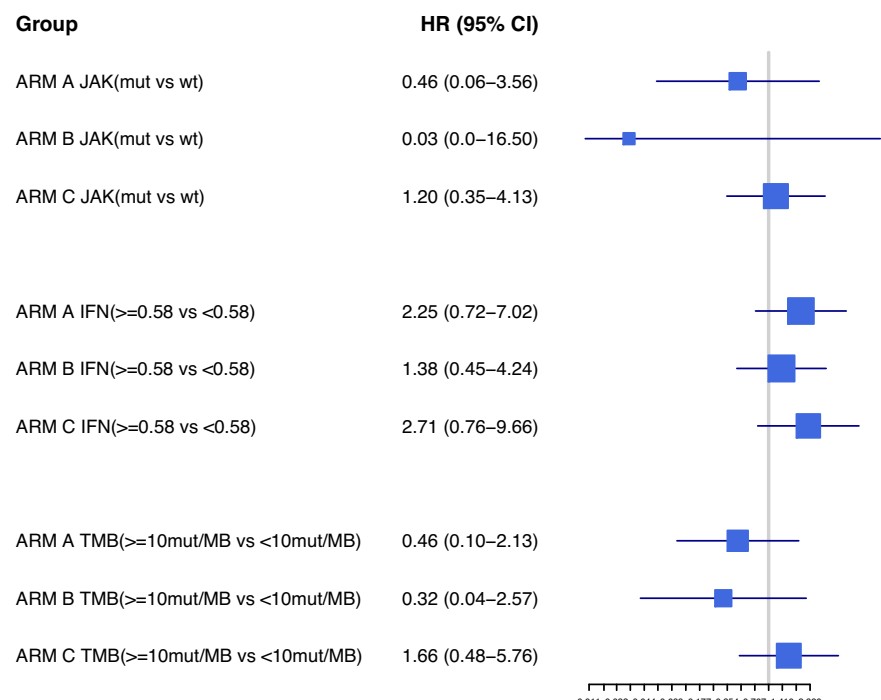

**Fig. 2 | HR according to biomarkers analysis.** Forest plot representing HR for Overall Survival according to Jak mutations, IFN gamma expression and TMB in Arm A (targeted therapy followed by immunotherapy), Arm B (immunotherapy followed by targeted therapy), and Arm C (a course of targeted therapy preceding immunotherapy and targeted therapy) of SECOMBIT. Source data are provided as a Source Data file.

was performed in samples from 29, 25, and 30 patients from Arms A, B, and C, respectively. The targeted NGS panel included a total of 409 cancer-related genes corresponding to 1.2-Mb of exonic sequence and 0.45-Mb of intronic sequence. Initially, molecules significantly correlated with outcomes were evaluated; the pathways in which these molecules are involved were then evaluated. TMB was directly calculated including variants at ≥5% allelic frequency at positions with ≥60× coverage. Serum levels of a panel of cytokines known to be involved in inflammation and anti-tumor immunity were quantified by at baseline in pre-treatment samples from 27 patients in arm A, 28 patients from Arm B, and 34 patients in Arm C (Fig. 2).

A total of 8 patients in Arm A, 8 patients in Arm B, and 12 patients in Arm C had tumors that were TMB-H (≥10 mut/Mb). The numbers of patients with TMB-L tumors across Arms A, B, and C, were 20, 17, and 18, respectively. Considering the effect of TMB on the entire cohort across all arms, the $p$ value was 0.44 and the $p$ value of the test for interaction was 0.24. Considering each arm individually, the 4-year OS rates for patients with TMB-H versus TMB-L tumors in Arms A and B were 75% (95% CI 45–100) vs 51% (95% CI 27–76) and 86% (95% CI 60–100) vs 59% (95% CI 35–82). By contrast, in Arm C, the 4-year OS rates were 58% (95% CI 30–86) for patients with TMB-H tumors versus 72% (95% CI 51–93) for patients with TMB-L tumors (Supplementary Fig. 4).

Deleterious mutations were identified in *JAK1, JAK2 or JAK3* in 5 patients in Arm A, 7 patients in Arm B, and 14 patients in Arm C. *JAK* was determined to be wild-type in 24, 18, and 16 patients in Arms A, B, and

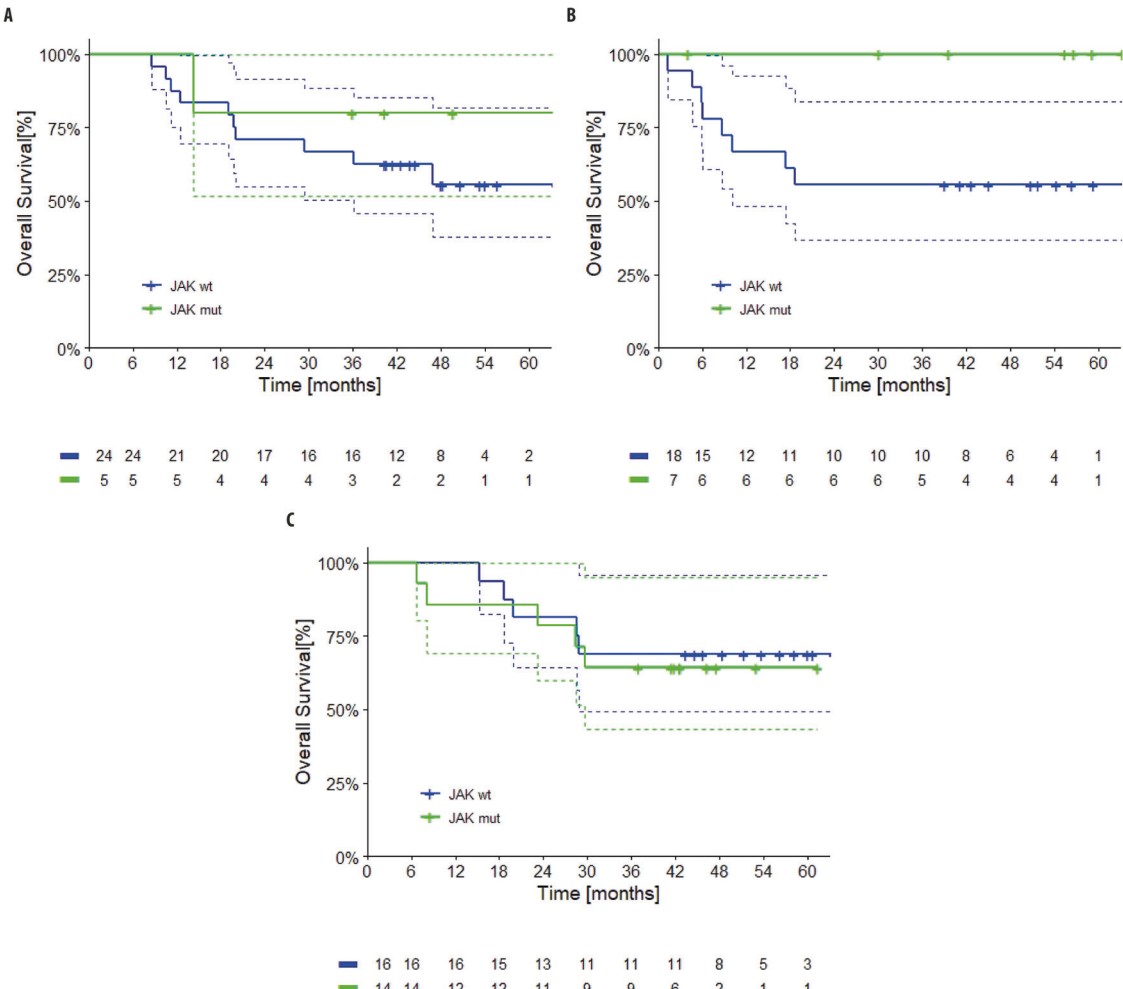

**Fig. 3 | Overall survival by *JAK* mutation status.** Kaplan–Meier survival curves for patients with wild type *JAK1/2* (blue) and deleterious mutations in *JAK1/2* (green) are shown, in Arm **A** (targeted therapy followed by immunotherapy), Arm **B** (immunotherapy followed by targeted therapy), and Arm **C** (a course of targeted therapy preceding immunotherapy and targeted therapy) of SECOMBIT. Dotted lines represent 95% confidence intervals. Source data are provided as a Source Data file.

C, respectively (Supplementary Table 2). Considering the effect of a deleterious *JAK* mutation across all arms, the p values was 0.19, with a p value of 0.72 for the test for interaction ($p < 0.10$ was considered as suggestive of difference). Within each arm individually, in Arms A and B, the presence of a deleterious *JAK* mutation was associated with numerically higher 4-year OS rates. The 4-year OS rates were 80% (95% CI 45–100) for those with a mutation vs 56% (95% CI 34–77) for those without for Arm A and 100% (95% CI n.e.) vs 56% (95% CI 33–78) for those with and without a mutation, respectively, in Arm B. None of the patients in Arm B whose tumors had deleterious *JAK* mutations died during the study. In Arm C, by contrast, the 4-year OS rates for patients with and without deleterious *JAK* mutations were 64% (95% CI 39–89) vs 69% (95% CI 46–92) (Fig. 3). TPFS followed a similar pattern to OS across the arms, with the 4-year TPFS rates for patients with versus without deleterious *JAK* mutations being 60% (95% CI 17–100) vs 39% (95% CI 18–60), 100% (95% CI n.e.) vs 50% (95% CI 25–75), and 64% (95% CI 39–89) vs 50% (95% CI 25–74) for Arms A, B, and C, respectively.

A total of 17, 16, and 23 patients in Arms A, B, and C, respectively, were identified as having high baseline serum IFNγ (cut-off was set at 0.58), based on a Receiver Operating Characteristic (ROC) analysis that identified a cut-off value of 0.580 pg/ml. The ROC analysis was based on patient status (alive/dead). The numbers of patients with low baseline serum IFNγ across the arms were 10, 12, and 11, in arms A, B, and C, respectively. The cut-off value for IFN was assessed through the

Youden's J index, which maximizes sensitivity and specificity in ROC curves, and confirmed by Maximally Selected Rank Statistics ("max-stat" package)

Considering IFNγ levels as a continuum, the HR for IFNγ and OS across all arms was 1.11 (95% CI: 0.95–1.30; $p = 0.20$) ($p < 0.10$ was considered as suggestive of difference). The *p* value for interaction between IFNγ and OS across all patients was 0.39. Within each arm, the 4-year OS rates for patients with high baseline IFNγ in Arms A and B were 29% (95% CI 8–51) and 50% (95% CI 25–75), respectively. Patients with low serum IFNγ at baseline in Arms A and B had 4-year OS rates of 52% (95% CI 16–89) and 58% (95% CI 30–86), respectively (HR: 1.93; 95% CI: 0.99–3.76). By contrast, in Arm C, the 4-year OS rate for patients with high IFNγ versus low IFNγ were 48% (95% CI 27–68) versus 73% (95% CI 46–99) (Fig. 4).

Serum levels of IFNγ were significantly associated with downstream cytokines and chemokines. Table 2 shows the correlations in the overall population. In Arm A, serum levels of IFNγ were positively correlated with serum IL-6, IL-10 and VEGFC. In ARM C, serum levels of IFNγ were positively correlated with CXCL10 and IL-17 (data not shown).

A hierarchically clustered co-correlational heatmap of cytokines and clinical variables in the overall population is provided in Fig. 5A, with levels of correlation between cytokines in each arm (Fig. 5B–D).

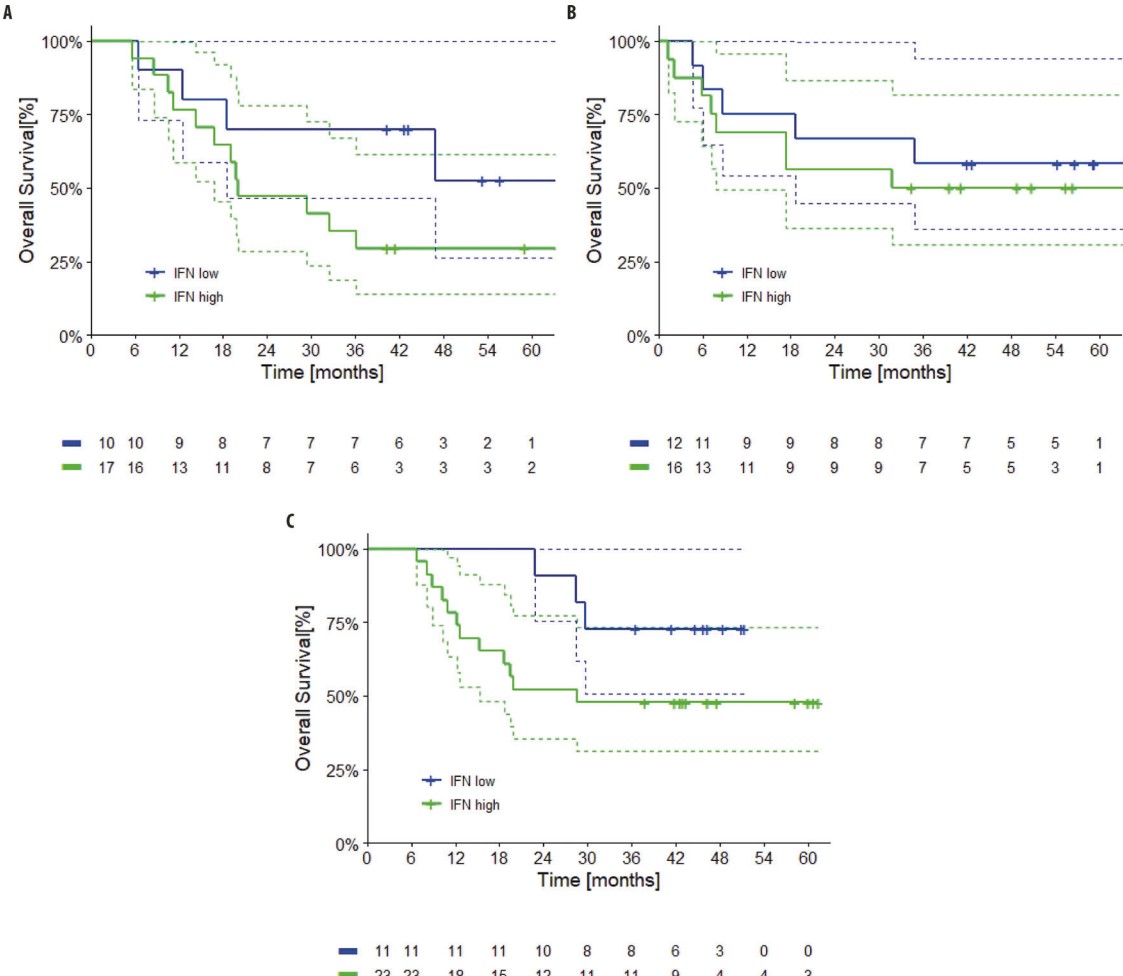

**Fig. 4 | Overall survival by baseline serum interferon gamma (IFNγ).**
Kaplan–Meier survival curves for patients with low baseline serum IFNγ (blue) and high baseline serum IFNγ (green) in Arm **A** (targeted therapy followed by immunotherapy), Arm **B** (immunotherapy followed by targeted therapy), and Arm **C** (a course of targeted therapy preceding immunotherapy and targeted therapy) of SECOMBIT. Dotted lines represent 95% confidence intervals. Source data are provided as a Source Data file.

Overall, MMP-9 and IFN-α were significantly more expressed in patients with SD or PD than in those with PR or CR (Supplementary Fig. 5).

## Discussion

These 4-year follow up data from the randomized, phase II SECOMBIT trial continue to demonstrate meaningful survival benefit with immunotherapy with or without an 8-week course of targeted therapy for the first-line treatment of *BRAF*-mutant metastatic melanoma. Furthermore, exploratory biomarkers analyses reveal unexpectedly improved 4-year OS rates for patients with defects in *JAK* as well as low baseline levels of serum IFNγ in the first-line immunotherapy arms. The correlation of all the other molecules represented in the panel with clinical outcomes was not significant.

Prior to 2021, no prospective data were available to inform selection of first-line therapy for *BRAF*-mutant metastatic melanoma. The results of SECOMBIT[13] and DREAMseq[14] established that combination immunotherapy should be considered the standard of care first-line regimen. Whether some patients may obtain additional benefit from a short course of targeted therapy before initiation of immunotherapy and biomarkers to select patients for the sandwich approach remained open questions.

The 4-year OS rates in Arms B and Arm C were 59% (95% CI 53–76) and 63% (95% CI 47–71), compared to 46% (95% CI 33–59) for Arm A.

In CheckMate 067, the 5-year OS rate was 60% in patients with *BRAFV600*-mutant tumors[4]. Long tails on the survival curves are now becoming evident in all three arms of the study, even in Arm A, reflecting the planned switch to immunotherapy after the first PD. As expected, the OS curves for Arms A and B cross, with the immunotherapy first approach outperforming the targeted therapy first approach after roughly the 1-year mark. Of note, the OS and TPFS curve for Arm C, the sandwich approach, remains above those for both other arms for the first roughly 18 months, after which the trajectory is nearly identical to Arm B. Strikingly, both early and long-term TPFS and OS benefit in Arm C was maintained in the subgroups of patients with disease features indicative of compromised immunity that are known to predict poor outcomes with checkpoint blockade, including elevated LDH[27–29], TMB-L[30], and low IFNγ[31–33].

Low serum levels of interferon at baseline as well as mutations in *JAK* were associated with improved 4-year survival in the sandwich and immunotherapy-first treatment arms. Strikingly, all patients with JAK mut in ARM B have 100% OS. These results seemingly conflict with well-characterized mechanisms of primary[20] and secondary resistance[19] to anti-PD-1, as well as the predictive role of tumor IFNγ-associated gene expression signatures in response to checkpoint blockade[31], although these studies focused on IFNγ signaling in tumor tissue as opposed to serum concentrations of the cytokine. Indeed, it cannot be ruled out that pleiotropic effects of IFNγ have a role in this event[21–26]. Loss of

**Table 2 | Correlation of serum levels of IFNy with downstream cytokines and chemokines in the whole study cohort**

| | Spearman correlation coefficient | Nominal P value | Adjusted P value* |
|---|---|---|---|
| VEGFC | 0.202 | 0.064 | 0.54 |
| TNFALPHA | 0.189 | 0.083 | 0.54 |
| IL-4 | −0.187 | 0.086 | 0.54 |
| IL-2 | −0.146 | 0.18 | 0.68 |
| PDGF_BB | 0.144 | 0.19 | 0.68 |
| IL-15 | 0.129 | 0.24 | 0.68 |
| IL1b_IL1F2 | 0.126 | 0.25 | 0.68 |
| MMP9 | 0.106 | 0.33 | 0.78 |
| ANGIOPIETIN2 | −0.082 | 0.46 | 0.94 |
| CCL2 | 0.074 | 0.50 | 0.94 |
| CXCL5 | 0.057 | 0.60 | 0.94 |
| PIGF | −0.053 | 0.63 | 0.94 |
| FGF | 0.038 | 0.73 | 0.94 |
| CXCL10 | 0.038 | 0.73 | 0.94 |
| HB | 0.089 | 0.74 | 0.94 |
| VEGF | 0.027 | 0.81 | 0.96 |
| IL-6 | −0.003 | 0.97 | 0.98 |
| IL-17 | 0.002 | 0.98 | 0.98 |
| IL-10 | 0.003 | 0.98 | 0.98 |

P values are adjusted by Benjamini–Hochberg method.

tumor antigenicity cannot account for the results, as 3 of the 7 patients in Arm B with *JAK*-mutant tumors were also TMB-L and 6 of the patients with low baseline serum IFNy were TMB-H. It has recently been shown that prolonged IFN stimulation promotes cancer cells resistance to checkpoint blockade by inducing epigenetic features of inflammatory memory[34]. Strikingly, TMB-H was not associated with improved survival in Arm C, despite corresponding to better outcomes in Arms A and B. The findings in Arm C are consistent with exploratory biomarkers analyses from COMBI-AD showing that high TMB is associated with reduced clinical benefit from adjuvant targeted therapy for resected stage IIIA (lymph node metastases >1 mm), IIIB, or IIIC cutaneous melanoma, especially if the IFNy signature is below the median[35]. In SECOMBIT, arm C received an initial administration of targeted therapy followed by immunotherapy. We hypothesize that this short initial administration of targeted therapy could modify some biological mechanisms underlying a resistance to immunotherapy.

Strikingly, all the patients with tumors with deleterious mutations in *JAK* in Arm B were still alive at extended follow up. These results seemingly conflict with the canonical role of *JAK1/2* mutations leading to resistance to anti-PD-1 therapy in melanoma[20], and the established mechanism of PD-L1 regulation by the type II interferon receptor singling pathway via JAK1 and JAK2[36]. However, in other highly immunogenic tumors such as microsatellite instability-high colorectal cancer, intact JAK signaling is or antigen presentation machinery is not required for outstanding outcomes with immunotherapy[37,38]. Improved responses to anti-PD-1 have even been reported in patients with *JAK*-mutant colorectal cancers[39]. The mechanisms underlying improved responses to anti-PD-1 in *JAK*-mutant tumors are still incompletely understood. Administration of TLR-9 agonists overcomes anti-PD-1 resistance in murine models[40], and a serendipitous activation of innate immunity via infection or other perturbation may have occurred in the patients with *JAK*-mutant tumors in our cohort. In lymphomas, chromosomal alterations in the region carrying *JAK* frequently cause overexpression of PD-L1[41], and oncogenic *JAK* upregulates PD-L1 in myeloproliferative neoplasms[42]. Whether the mutations predicted to be damaging by the PolyPhen2 scores we detected in this study might also alter the transcriptional regulation of PD-L1 is not known, however, the close link between the JAK/STAT signaling pathway and PD-L1 as illustrated by a network interaction map of direct and functional protein-protein interactions derived from the STRING database (https://string-db.org/) in Fig. 6 supports such a possibility.

IFNy signaling is broadly conserved in melanoma cells[43], and attributed as the main driver of response to PD-1 blockade[32]. Binding of IFNy to the IFNGR1/2 complex results in activation of JAK1 and JAK2, phosphorylation and nuclear translocation of STAT3, and transcriptional activation of primary and secondary interferon-responsive genes[44]. In tumor cells, IFNy causes cell cycle arrest and upregulation of the antigen presentation machinery, while lymphocytes and endothelial cells exposed to the cytokine secrete chemoattractants such as CXCL10[32,43,44]. IFNy signaling is well-known to be a double-edged sword, however, that both enhances and inhibits anti-tumor immunity[44,45]. Effector T cell activity in the TME induces adaptive immune resistance mechanisms[46] including IFNy-mediated upregulation of immune checkpoints, including PD-L1[47]. *BRAF*V600 further enhances IFNy-inducible PD-L1 expression by enhancing translation via a STAT1-dependent mechanism[48]. Consistent with this, in Arm A, serum IFNy was associated with significant changes in IL-6, IL-10, and VEGF, indicative of an immunosuppressed TME.

IFNy signaling in tumor cells leads to direct suppression of lymphocyte effector functions via multiple checkpoints and inhibitory pathways beyond PD-L1, all of which may have contributed to the survival outcomes across the arms. In particular, BRAF inhibition, which all patients in Arm C received prior to ipilimumab plus nivolumab treatment, decreases IFNy-stimulated PD-L1 expression while enhancing expression of the immunosuppressive lectin Gal-1[49]. Additionally, IFNy causes the upregulation of non-canonical MCH class I molecules such as human leukocyte antigen (HLA)-G and HLA-E[50], which limit anti-melanoma cytotoxicity by T cells[51]. Multiple checkpoint ligands are upregulated on tumor cells by IFNy, including Qa-1b, which binds NKG2A/CD94 on NK cells and activated CD8+ effector T cells[22] and CD155, which interacts with the T cell immunoreceptor with Ig and ITIM domains (TIGIT)[52]. Adaptive immune evasion has long been known to involve IDO upregulation and Treg differentiation as direct consequences of IFNy in the TME[46]. IFNy also indirectly limits T cell effector function by causing a switch toward tolerogenic IDO+ dendritic cells, which support Treg differentiation[53], contribute to MSDC recruitment[54], and impair CD8+ T cell priming[24].

Furthermore, the role of IFNy signaling in response to anti-PD-1 has mainly been characterized in the context of monotherapy. In SECOMBIT, patients were treated with combination anti-PD-1 and anti-CTLA-4. Dual checkpoint blockade, distinct from anti-PD-1 alone, leads to an expansion of activated terminally differentiated effector CD8+ T cells[55]. In mice with low tumor burden, IFNy signaling has been demonstrated to cause clonal deletion of tumor antigen-specific T cells upon dual checkpoint blockade due to activation-induced cell death[25]. Non-lymphocyte populations also play a role in suppressing cytotoxic activity by effector T cells. IFNy also promotes immune tolerance via cross-presentation of tumor antigens by lymphatic endothelial vessels, which enhances Treg function leading to apoptosis of antigen-specific CD8+ T cells in the draining lymph node[21,56,57]. While patients in SECOMBIT all had advanced melanoma, the short course of BRAF inhibition in Arm C may have shifted the immune infiltrate toward dominance of newly activated effector T cells by normalizing the vasculature, debulking the tumor, increasing glucose availability[58,59], and alleviating hypoxia[60]. Patients with elevated IFNy at baseline in Arm C may have then experienced hyperactivation and apoptosis of antigen-specific effector T cells upon ipilimumab plus nivolumab treatment. Further supporting this model, TMB-H did not associate with improved survival in Arm C, indicating that highly antigenic tumors do not confer a survival advantage for patients treated with the sandwich approach. Serum IFNy in Arm C also positively correlated

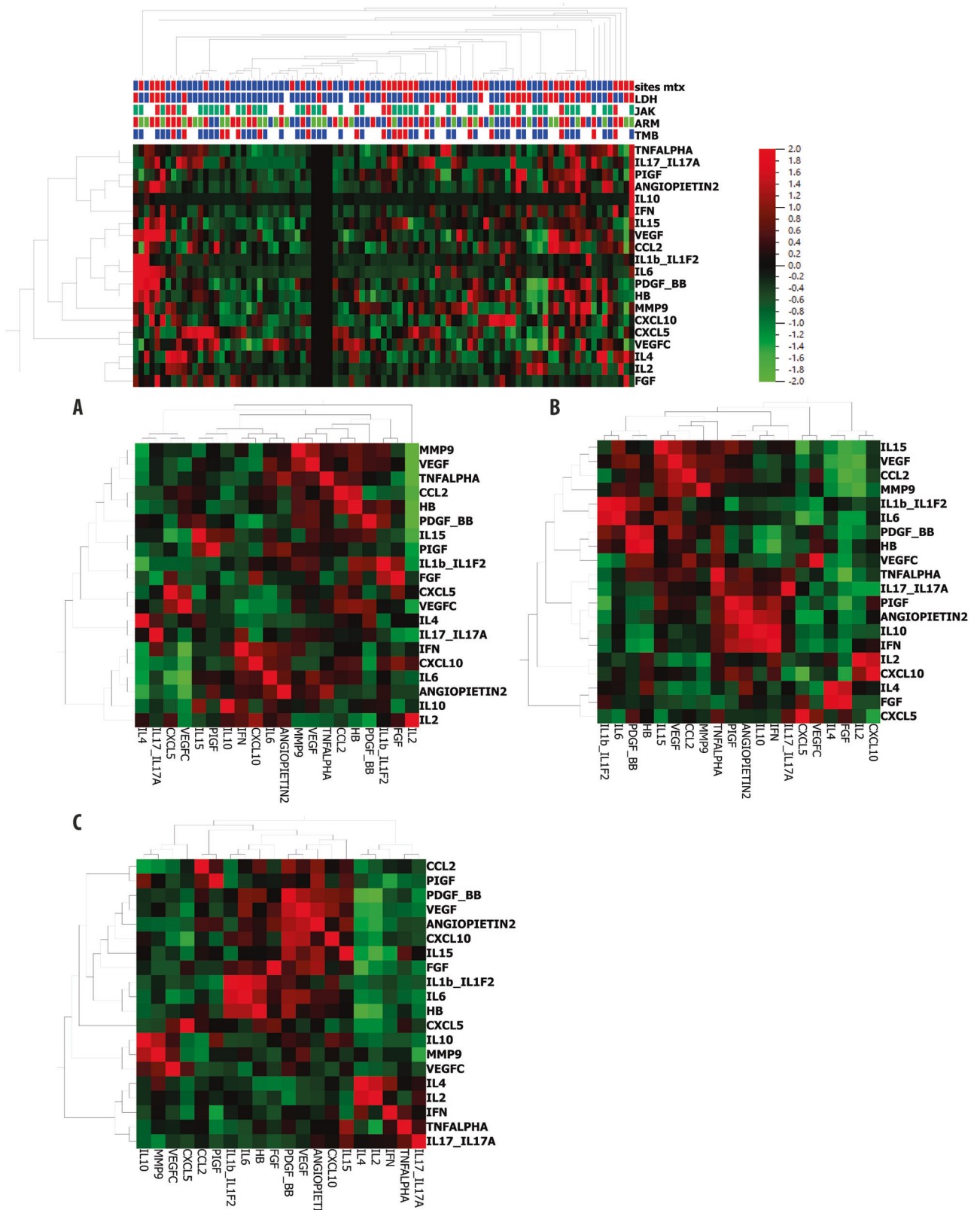

**Fig. 5 | Heatmaps.** Panel **A** (*n* = 89 patients) shows hierarchical co-clustering of clinical variables and baseline serum cytokine levels in the overall cohort. Levels of correlation of each cytokine with the other ones in arm A (panel **B**, *n* = 27 patients), in arm B (panel **C**, *n* = 28 patients), in arm C (panel **D**, *n* = 34 patients). The setting for the visible lower and upper scale bounds is two standard deviations (ANOVA was used for comparison of groups). *shows unavailable cytokines.

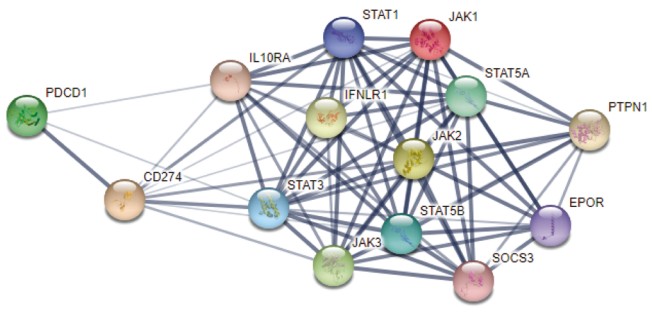

**Fig. 6 | Network interaction map depicting regulatory interactions between the JAK/STAT signaling pathway and the PD-1(PDCD1)/PD-L1(CD274) axis.** Line thickness represents the strength of data. STRING (https://string-db.org/) uses a spring model to generate the network images. Nodes are modeled as masses and edges as springs; the final position of the nodes in the image is computed by minimizing the 'energy' of the system[63,64].

with CXCL10 and IL-17, further underscoring that a lack of T cell infiltration into the tumor was not limiting for outcomes in the group of patients with high baseline serum IFNγ treated with the sandwich approach.

Additional analyses are needed to confirm the trends that we observed and to definitively establish the mechanisms underlying the survival outcomes. Limitations of this study include the small number of patients included in the biomarkers analysis as well as the lack of high-resolution information on cellular populations and cytokine levels within the TME. Our analyses solely focused on serum IFNγ. While IFNγ-associated gene expression profiles in tumor samples are well-established predictors of response and resistance to checkpoint blockade[19,20,31], the degree of correlation between serum and intratumoral levels of IFNγ in melanoma is not well established. Peripheral IFNγ has been shown to predict responses to therapeutic vaccines in melanoma[61], but it has not yet been validated as an independent biomarker of response to checkpoint blockade. The correlation we observed between serum IFNγ and downstream cytokines supports the physiological relevance to tumor biology in our study. Our data indicate that low baseline serum IFNγ may have some utility to select patients for BRAF/MEK induction before ipilimumab plus nivolumab. However, a multi-factor biomarker involving several cytokines or multiple clinical and laboratory features will likely be necessary to inform treatment decisions.

In conclusion, the 4-year survival results from SECOMBIT further cement the status of immunotherapy as the preferred first-line treatment option for *BRAF*V600-mutant metastatic melanoma. Some patients with significantly impaired immunity at baseline may require brief course of BRAF/MEK inhibition to halt rapidly progressing disease and prime the TME for unrestrained CD8+ T cell effector function with checkpoint blockade. Our results also are hypothesis-generating for further investigation into peripheral cytokine levels at baseline as predictors of benefit with immunotherapy. Further studies are needed to validate biomarkers for patient selection and elucidate the molecular mechanisms responsible for durable clinical benefit with sequential combination immunotherapy and targeted therapy.

## Methods
### Study design
This study was designed in 2015 as a phase II, open-label randomized trial with no formal comparative test and a single-stage design for each arm. Patients were enrolled at 37 academic medical centers in 9 countries. The trial protocol was approved by the appropriate ethics body at each participating institution and is available in the Supplementary Information file. An independent data monitoring committee oversaw the trial. SECOMBIT is registered at ClinicalTrials.gov

(NCT02631447). The study design and conduct complied with all current regulations regarding the use of human study participants and was conducted in accordance with the criteria set by the Declaration of Helsinki.

### Participants
Participants aged ≥18 years and Eastern Cooperative Oncology Group (ECOG) performance status (PS) 0 or 1 with histologically confirmed unresectable stage III or stage IV melanoma with measurable disease by computed tomography (CT) or Magnetic Resonance Imaging (MRI) per RECIST 1.1 criteria[62] and tumors harboring a *BRAFV600* mutation were enrolled. All patients provided written informed consent before enrollment. Detailed eligibility criteria were published previously[13] and are available in the study protocol (available in the Supplementary Information file). The first patient was enrolled on December 23rd 2016, and the last one on May 23rd 2019.

### Randomization
Patients were randomized 1:1:1 across treatment arms. Arm A received encorafenib plus binimetinib until progressive disease [PD], followed by ipilimumab plus nivolumab until second PD. Arm B received ipilimumab plus nivolumab until PD followed by encorafenib plus binimetinib until second PD. Arm C ('sandwich' or 'induction/maintenance') received encorafenib plus binimetinib for 8 weeks followed by ipilimumab plus nivolumab until PD followed by encorafenib plus binimetinib until second PD). Patients were stratified by number of involved tumor sites and LDH elevation (IIIb/c – M1a – M1b, M1c with LDH ≤ 2ULN, and M1c with elevated LDH > 2 ULN).

### Procedures
Patients were treated with encorafenib plus binimetinib (encorafenib at 450 mg orally once daily, binimetinib at 45 mg orally twice daily) and ipilimumab plus nivolumab (ipilimumab 3 mg/kg, nivolumab 1 mg/kg once every 3 weeks for 4 cycles, followed by nivolumab 3 mg/kg once every two weeks) according to the treatment sequence for each arm. Tumor responses were assessed by investigators every 8 weeks for the first year and every 12 weeks thereafter while on study according RECIST version 1.1[62]. Survival rates at 4 years were estimated using Kaplan–Meier methods. Tumor tissue from an unresectable or metastatic site of disease was required per protocol to be collected for biomarker analyses at baseline and at progression, as a prespecified analysis. Peripheral blood was collected at baseline. A more detailed description of assessments is available in the trial protocol (available in the Supplementary Information file).

### Statistical analyses
PFS was calculated as the time between randomization and evidence of relapse or death, whichever occurs first or censored at the time of last evaluation. OS was calculated as the difference between randomization and death, or censored at the time of last follow-up. The Kaplan–Meier method was used to estimate OS and PFS. Hazard Ratios and their 95% confidence intervals were calculated using the Cox regression model. Associations among cytokines were evaluated with the Spearman coefficient and P values were adjusted for multiplicity using the Benjamini–Hochberg procedure. Mann–Whitney test was used to evaluate the expression of cytokines in subgroups of patients. Statistical analysis was performed using IBM-SPSS version 21.0 or later and R v.4.02 on a Windows 10 operating system.

### FFPE DNA extraction and next generation sequencing
Genomic DNA was isolated from formalin-fixed and paraffin-embedded (FFPE) tissue sections from melanoma patients, using the GeneRead DNA FFPE Kit (Qiagen, Hilden, Germany) and following manufacturer´s instructions. DNA concentrations were assessed by Qubit 2.0 Fluorometer with Qubit dsDNA HS (High Sensitivity) Assay

Kit (Life Technologies, Carlsbad, CA, USA). Next generation sequencing (NGS) analyses were performed using the Ion GeneStudio S5 System with the Oncomine Tumor Mutational Load panel (OTML) that includes a total of 409 cancer-related genes arranged in two primer pools. The total genomic space splits up into a 1.2-Mb exonic region and a 0.45-Mb intronic region. Libraries were generated starting from 10 ng of DNA per primer pool for a total of 20 ng of input DNA using the Ion AmpliSeq Library Kit Plus, barcoded with Ion Xpress Barcode Adapters (Life Technologies) and purified with Agencourt Ampure XP Beads (Beckman Coulter Life Sciences, Indianapolis, USA). The PCR amplicons were diluted to a final concentration of 70 pM and pooled together; emulsion PCR and Chip loading steps were performed by the Ion Chef Instrument. Libraries sequencing was performed loading four samples on each Ion 540 chip. Raw sequence data were analyzed with the Torrent Suite™ Software (Version 5.10.2). Torrent Mapping Alignment Program was used to map reads against hg19 human reference genome. Torrent Variant Caller Plugin (V 5.10.1.19) and Coverage Analysis and were used to perform initial quality control and to assess amplicon coverage for regions of interest. The resulting BAM (variant call format, VCF) files were transferred to Ion Reporter software version 5.16 (Thermo Fisher Scientific) for secondary analyses, including variants annotation and Tumor Mutational Burden (TMB) calculation. We estimated the proportion of sequence reads that matched a particular DNA variant by dividing this count by the overall number of reads at the relevant genomic locus in order to obtain the Variant Allele Fraction (VAF) for each somatic variant recovered from VCF files. The following formula was used by us:

VAF = (N_ref + N_var)/N_var; By counting reads that supported both the reference allele (N_ref) and the variant alleles (N_var) at the pertinent genomic region, we were able to determine the results.

### Variant calling and TMB assessment
Variants were annotated using the following databases: 5000Exomes Global MAF (V 2016_11_08), ClinVar (V 2020_11_21), COSMIC (V 92), dbSNP (V 154), DGV (V 2020_02_25), DrugBank (V 2020_10_29), Gene Ontology (V 2020_11_18) OMIM (V 2020_12-02), Pfam (V 33), PhyloP Scores (V 2016_09-19). Bulk analyses were carried out to obtain a total amount of at least 10 mutated alleles for each candidate variant, according to the following minimum criteria: coverage of ≥200 reads and frequency of mutated alleles ≥5%. For specific gene pathways (*BRAF-NRAS, PI3K-PTEN*, *JAK*1/2/3-CTNNB1), we performed a more accurate analysis with less stringent parameters: at least 5 mutated alleles for each candidate variant with a coverage of ≥100 reads. Mapped reads and variant calls were visualized using Integrative Genome Viewer (IGV). TMB (Algorithm Version 4.0) value was directly calculated by the Ion Reporter™ Software including variants at ≥5% allelic frequency at positions with sufficient read coverage (≥60).

### *BRAF* V600 ddPCR detection
Samples in which V600 BRAF mutation was not identified by NGS technology were subjected to a more sensitive analysis by Droplet Digital PCR (ddPCR) assay. Briefly, 10 ng of DNA quantified with Qubit dsDNA HS were used to screen V600E/K/R mutations of the BRAF gene with the QX200 Droplet Digital PCR System (Bio-Rad). The Master mix for ddPCR included 11 μL of 2X ddPCR Supermix for Probes (no dUTP, Bio-Rad), 1 μL of 20X BRAF V600 Screening Assay (#12001037, Bio-Rad) and 10 μL of DNA for a final volume of 22 μL. 20 μl of reaction master mix was added to the DG8 cartridges (Bio-Rad, 1864008) with the addition of 70 μl Droplet Generation Oil for Probes (Bio-Rad, 1863005); QX200 Droplet Generator (Bio-Rad, 10031907) was used to produce droplet emulsion. Droplets were PCR amplified on a T100 Touch thermal cycler (Bio-Rad, 1861096) with the following program: 95 °C for 10′, 40 cycles of 94 °C for 30″ and 55 °C for 1′, 98 °C for 10′ and 4 °C 30′, with a ramp rate of 2.5 °C/s. Readout of droplet fluorescence

was performed by the Droplet Reader QX200 Droplet Reader (Bio-Rad, 1864003) and analyzed with the QuantaSoft Analysis Pro Software Version 1.7.4 (Bio-Rad, USA).

### Polyphen score for *JAK* mutation characterization
The PolyPhen-2 score predicts the possible impact of an amino acid substitution on the structure and function of a human protein. This score represents the probability that a substitution is damaging. For each variant, Ion Reporter™ Software reporting the pph2-prob PolyPhen-2 score was used at https://ionreporter.thermofisher.com/ir/secure/analyses.html. The PolyPhen-2 score ranges from 0.0 (tolerated) to 1.0 (deleterious). Variants with scores of 0.0 are predicted to be benign. Values closer to 1.0 are more confidently predicted to be deleterious. Scores of 0.0 to 0.15 were predicted to be benign, scores of 0.15 to 1.0 were predicted as possibly damaging, and scores of 0.85 to 1.0 were predicted confidently to be damaging. Any additional information on the IonReporter software version used can be find at https://assets.thermofisher.com/TFS-Assets/LSG/manuals/MAN0019148_IonReporter_5_16_UG.pdf.

### Cytokine analysis
Patients' baseline peripheral blood was collected into serum tubes. Serum was collected by centrifugation at 1700 × *g* for 10 min and aliquots were immediately stored at −80 °C until use.

A panel of 22 inflammatory cytokines and molecules was quantified using a Luminex platform (Human Cytokine Discovery, R&D System, Minneapolis, MN) for the simultaneous detection of the following molecules: IL-6, IL-15, IL-17, IFN-γ, TNFα, CCL-2, CXCL5, CXCL10, ANG-2, FGF2, HB-EGF, VEGF, VEGF-C, MMP9, PDGF-BB, PIGK and by High sensitivity ELISA for IL-1β, IL-2, IL-4, IL-10, IL-12p70, were evaluated at baseline from 93 patients following the manufacturer's instruction. For each sample, two technical replicates were performed.

## Data availability
All data generated in this study have been deposited in Zenodo (https://zenodo.org/records/8386539), the variant allele frequency (VAF) data have been deposited in the European Variation Archive (EVA) at EMBL-EBI under the accession number PRJEB70957). The accessions associated with the submission are: Project: PRJEB70957, Analyses: NGS Analysis = > ERZ22145378. Accession is public in both databases. Anonymous characteristics of patients at baseline are shared. Clinical values and biomarkers reported in the manuscript are shared. The study protocol is available as Supplementary Note in the Supplementary Information file. The remaining data are available within the Article, Supplementary Information or Source Data file. Source data are provided with this paper.

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

## Acknowledgements

The authors thank the patients and families who made this trial possible. Additionally, the authors acknowledge the clinical study teams and CRO who participated in the trial and in particular Paola Schiavo e Mariarita Arenella from CRT (Clinical Research Technology - Salerno). This study was supported by unconditioned grants from Bristol-Myers Squibb (Princeton, NJ) and Array Biopharma Inc./Pfizer (Boulder, CO). Sam Million-Weaver, PhD provided medical writing support. The authors also thank the participating investigators who did not enroll any patients and thus are not included as authors on the paper, Koelblinger P, Hafner C, Hoeller C (Austria), Weide B (Germany), Larkin J, Lorigan P (UK). This study was supported by unconditioned grants from Bristol-Myers Squibb (Princeton, NJ) and Array Biopharma Inc./Pfizer (Boulder, CO).

## Author contributions

Conception and design: P.A.A., Mi. G., M.T., I.M., A.M.G., R.D., V.C.S. Administrative support: M.P., R.D. Provision of study materials or patients: M.M., P.F.F., Ma. G., P.R., A.C., V.F., A.A., E.M., H.G., E.R., M.T.F., H.H., F.S., M.D.V., M.G.C., M.F.S., D.M., M.P., I.M., V.C.S., P.Q.; Collection and assembly of data: P.A.A., M.M., P.F.F., A.C., Ma. G., P.R., V.F., H.G., E.R., M.T.F., C.L., H.H., F.S., M.D.V., M.G.C., M.F.S., D.M., M.P., J.B., R.D., V.C.S., P.Q.; Data analysis and interpretation: P.A.A., P.F.F., V.F., A.A., A.C., E.M., H.G., C.L., H.H., M.D.V., M.G.C., A.M.M., S.D.P., M.F.S., D.M., I.M., G.P., D.G., R.D., V.C.S., M.C., M.P., C.P., L.P., M.G.V. Manuscript writing: All authors. Final approval of manuscript: All authors. Accoun-table for all aspects of the work: All authors.

## Competing interests

The authors declare the following competing interests: P.A.A.: Stock and Other Ownership Interests: PrimeVax. Consulting or Advisory Role: Bristol Myers Squibb, Roche/Genentech, Merck Sharp & Dohme, Novartis, Array BioPharma, Merck Serono, Pierre Fabre, Incyte, Med-Immune, AstraZeneca, Sun Pharma, Sanofi, Idera, Ultimovacs, Sandoz, Immunocore, 4SC, Alkermes, Italfarmaco, Nektar, Boehringer Ingel-heim, Eisai, Regeneron, Daiichi Sankyo, Pfizer, OncoSec, Nouscom, Takis Biotech, Lunaphore Technologies, Seattle Genetics, ITeos Therapeutics, Medicenna, Bio-Al Health, ValoTx. Research Funding: Bristol Myers Squibb (Inst), Roche/Genentech (Inst), Array BioPharma (Inst), Sanofi (Inst), Pfizer (Inst). Travel, Accommodations, Expenses: Merck Sharp & Dohme, Pfizer. M.M. Honoraria: MSD Oncology, Novartis, Pierre Fabre, Sanofi/Aventis, Bristol Myers Squibb/Sanofi. Consulting or Advisory Role: Bristol Myers Squibb, MSD Oncology, Novartis, Pierre Fabre. Research Funding: Novartis (Inst). Pier Francesco Ferrucci. Expert Testi-mony: Delcath Systems. M.G. Consulting or Advisory Role: BMS, Novartis, Pierre Fabre. Speakers' Bureau: BMS, Novartis, Pierre Fabre. Research Funding: MSD. P.R. Honoraria: Bristol Myers Squibb, MSD, Novartis, Roche, Pfizer, Pierre Fabre, Sanofi, Merck. Consulting or Advisory Role: Novartis, Blueprint Medicines, Bristol Myers Squibb, Pierre Fabre, MSD, Amgen. Speakers' Bureau: Pfizer, Novartis, Pierre Fabre. Research Funding: Novartis (Inst), Roche (Inst), Bristol Myers Squibb (Inst). Travel, Accommodations, Expenses: Orphan Europe, Pierre Fabre. V.F. Consulting or Advisory Role: Bristol Myers Squibb, Novartis. Speakers' Bureau: Bristol Myers Squibb, Novartis, Pierre Fabre, MSD Oncology. A.A. Consulting or Advisory Role: BMS, Roche, Novartis, Pierre Fabre, MSD, Merck, Sanofi. Speakers' Bureau: Pierre Fabre, Novartis, MSD, BMS, Roche, Merck, Sanofi. Research Funding: Pierre Fabre (Inst), Novartis (Inst), Roche (Inst), BMS (Inst), MSD (Inst), Merck (Inst), Sanofi (Inst). Travel, Accommodations, Expenses: BMS, MSD, Novartis, Pierre Fabre. H.G. Honoraria: Bristol Myers Squibb, MSD Oncology, Pierre Fabre, Sanofi/Regeneron. Consulting or Advisory Role: Bristol Myers Squibb, MSD Oncology, Amgen, Pierre Fabre, Sanofi/Regeneron. Research Funding: Bristol Myers Squibb (Inst), Roche (Inst), MSD Oncology (Inst), Amgen (Inst), Novartis (Inst), Iovance Biother-apeutics (Inst). Travel, Accommodations, Expenses: Bristol Myers Squibb, MSD, Amgen, Pfizer. E.R. Honoraria: Amgen, Bayer, Bristol Myers Squibb, Merck Sharp Dohme, Merck, Novartis, Pierre Fabre, Roche, Sanofi. Consulting or Advisory Role: Amgen, Bayer, Bristol Myers Squibb, Merck Sharp & Dohme, Merck, Novartis, Pierre Fabre. Speakers' Bureau: Amgen, Bristol Myers Squibb, Merck Sharp & Dohme, Merck, Novartis, Pierre Fabre, Sanofi. Research Funding: Amgen (Inst), Bristol Myers Squibb (Inst), Merck Sharp & Dohme (Inst), Novartis (Inst), Pierre Fabre (Inst), Roche (Inst), Cure. The remaining authors declare no other competing interests.

## Additional information

Paolo A. Ascierto [1] ✉, Milena Casula[2], Jenny Bulgarelli [3], Marina Pisano[2], Claudia Piccinini[3], Luisa Piccin[4], Antonio Cossu[5], Mario Mandalà [6,7], Pier Francesco Ferrucci [8], Massimo Guidoboni [3], Piotr Rutkowski [9], Virginia Ferraresi[10], Ana Arance [11], Michele Guida[12], Evaristo Maiello[13], Helen Gogas[14], Erika Richtig[15], Maria Teresa Fierro[16], Celeste Lebbe [17], Hildur Helgadottir[18], Paola Queirolo[19,20], Francesco Spagnolo[19], Marco Tucci[21], Michele Del Vecchio[22], Maria Gonzales Cao [23], Alessandro Marco Minisini[24], Sabino De Placido[25], Miguel F. Sanmamed [21], Domenico Mallardo [1], Miriam Paone[1], Maria Grazia Vitale[1], Ignacio Melero [26], Antonio M. Grimaldi[1,27], Diana Giannarelli[28], Reinhard Dummer [29], Vanna Chiarion Sileni [4,30] & Giuseppe Palmieri [2,30]

[1]Department of Melanoma, Cancer Immunotherapy and Development Therapeutics. I.N.T. IRCCS Fondazione "G. Pascale", Napoli, Italy. [2]Immuno-Oncology & Targeted Cancer Biotherapies, University of Sassari - Unit of Cancer Genetics, IRGB-CNR, 07100 Sassari, Italy. [3]Immunotherapy, Cell Therapy Unit and Biobank Unit, IRCCS Istituto Romagnolo per lo Studio dei Tumori (IRST) "Dino Amadori", Meldola, Italy. [4]Melanoma Oncology Unit, Veneto Institute of Oncology IOV-IRCCS, Padova, Italy. [5]Department of Medicine, Surgery and Pharmacy, University of Sassari, Sassari, Italy. [6]University of Perugia, Perugia, Italy. [7]Department of Oncology and Haematology, Papa Giovanni XXIII Cancer Center Hospital, Bergamo, Italy. [8]Biotherapy of Tumors Unit, Department of Experimental Oncology, European Institute of Oncology, IRCCS, Milan, Italy. [9]Department of Soft Tissue/Bone Sarcoma and Melanoma, Maria Sklodowska Curie National Research Institute of Oncology, 02-781 –, Warsaw, Poland. [10]Department of Medical Oncology 1, IRCCS Regina Elena National Cancer Institute, Rome, Italy. [11]Department of Medical Oncology, Hospital Clínic Barcelona, 08036 Barcelona, Spain. [12]Rare Tumors and Melanoma Unit, IRCCS Istituto dei Tumori "Giovanni Paolo II", Bari, Italy. [13]Oncology Unit, Foundation IRCCS Casa Sollievo della Sofferenza, San Giovanni Rotondo, Italy. [14]First Department of Medicine, National and Kapodistrian University of Athens, Athens, Greece. [15]Department of Dermatology, Medical University of Graz, Graz, Austria. [16]Department of Medical Sciences, Dermatologic Clinic, University of Turin, Turin, Italy. [17]Dermato-Oncology and CIC AP-HP Hôpital Saint Louis,Cancer Institute APHP. Nord-Université Paris Cite F-75010, Paris INSERM U976, France. [18]Department of Oncology-Pathology, Karolinska Institutet and Karolinska University Hospital Solna, Stockholm, Sweden. [19]Skin Cancer Unit, IRCCS Ospedale Policlinico San Martino, Genova, Italy. [20]Division of melanoma Sarcoma and Rare Tumors, IRCCS European Institute of Oncology, Milan, Italy. [21]Department of Interdisciplinary Medicine, Oncology Unit, University of Bari "Aldo Moro", Bari, Italy. [22]Unit of Melanoma Medical Oncology, Department of Medical Oncology and Hematology, Fondazione IRCCS Istituto Nazionale dei Tumori, Milan, Italy. [23]Department of Medical Oncology, University Hospital Dexeus, Barcelona, Spain. [24]Academic Hospital "Santa Maria della Misericordia", Udine, Italy. [25]Department of Clinical Medicine and Surgery, University of Naples "Federico II", Naples, Italy. [26]Department of Immunology and Oncology, Clínica Universidad de Navarra, Pamplona, Spain. [27]Medical Oncology Unit, AORN San Pio, Benevento, Italy. [28]Fondazione Policlinico Universitario A. Gemelli, IRCCS – Facility of Epidemiology and Biostatistics, Rome, Italy. [29]Department of Dermatology, University and University Hospital Zurich, Zurich, Switzerland. [30]These authors contributed equally: Vanna Chiarion Sileni, Giuseppe Palmieri. ✉e-mail: p.ascierto@istitutotumori.na.it

