## [Peer Review File · Nature Communications]

Sequential immunotherapy and targeted therapy for metastatic BRAF V600 mutated melanoma: 4-year survival and biomarkers evaluation from the phase II SECOMBIT trialEditorial Note: This manuscript has been previously reviewed at another journal that is not operating a transparent peer review scheme. Mentions of the previous journal have been redacted. This document only contains reviewer comments and rebuttal letters for versions considered at *Nature Communications*.

REVIEWER COMMENTS

Reviewer #2 (Remarks to the Author): ([Original journal name redacted] original Reviewer #2)

Ascierto et al have now improved their manuscript significantly in response to the concerns raised in the original review and provide important new data on both survival updates of the trial and exploratory biomarkers that differentiate response. However, I have a few remaining questions and recommendations, focused both in statistics, presentation, and in the exploratory biomarker analysis.

I clearly appreciate the bind the authors are in – the study was not designed or powered to compare the different arms, or further to do subgroup analyses, and thus analysis of differences between arms and in subgroups is exploratory as is now clear in the text. However, the results as presented are quite challenging for the reader to parse through, and statements in the discussion, e.g. “Low serum levels of interferon at baseline as well as mutations in JAK were associated with improved 4-year survival in the sandwich and immunotherapy-first treatment arms” require some type of (statistical) evidence to support the statement. I have two suggestions

- Inference of effect size and confidence intervals without p-values: a Cox PH analysis to generate estimated hazard ratios (as done in their JCO paper reporting earlier results) and CI would be helpful.

- A forest plot would help summarize results in overall and (biomarker-driven) subgroups, e.g. Figure 3 from [https://doi.org/10.1016/S0140-6736\(22\)00562-1](https://doi.org/10.1016/S0140-6736(22)00562-1)

Other specific questions/suggestions below:

- My preference would be to move TPFS survival curves from supplementary figures to main figures (e.g. Fig 1a and b)
- If you provide p-values/comparison, a clear description of the comparison and test performed should be provided (e.g. "...(a log-rank (?) test of difference in survival between TMB-H vs TMB-L tumors across all arms)...", lines 186-187; similarly in lines 195-196).
- Line 217 –,"The ROC analysis was based on patient status (alive/dead)." Alive or dead at what time point (4 years?)? How were censored patients considered?
- I appreciate that the authors have now provided the PolyPhen-2 score and an explanation of the interpretation, as well as JAK1/2/3 mutations from Arm B in Table S2. What is still not clear is whether the authors classified > a score of 0.15 ("possibly damaging") or > a score of 0.85 ("confidently damaging") as "deleterious mutations", or some other threshold. What should also be cited as a limitation is that the NGS is tumor-only targeted panel sequencing, which we know results in false positive calls (e.g. germline SNPs despite large germline filtering datasets without a matched normal to filter against) and systematically higher TMB (e.g. DOI: 10.1200/PO.19.00171).

The patient-level biomarker and outcome information is greatly appreciated and will be very helpful.

- Is there interferon gamma information as well? (specific levels would be best)
- A dictionary would be helpful (e.g. what is the column "PD" vs "Progr total")
- Notes about missing values/excluded patients would be helpful
- Data to be able to reproduce Fig 4 would be terrific.
- Is there age, gender, stage information that would be available?

I continue to be confused about the inference of Table 2, which shows differential correlation between ifn-gamma and other cytokines between arms. If there are indeed

differences between cytokine correlations between randomized arms before treatment, how do the authors explain/interpret this? Given that these are randomized arms and before treatment, my suggestion would be to examine the cytokine co-correlations across the whole cohort rather than cite differences between arms. This may result in more stable/clear clusters of cytokines that may provide insight into the relevant immune states (e.g. both CXCL10 and IL10 have a positive correlation with IFN-gamma in both arms cited, though only one is statistically significant; IL1b_IL1F2 is not statistically significantly associated with IFN-gamma but is positively associated in both arms and may be associated when considering the entire cohort)

In my mind, there are two distinct questions:

1) Are there specific immune states reflected by cytokine profiles that cluster (e.g. IFN-G, CXCL10, IL-10 are co-correlated)? (The co-correlational clustered heatmap which I suggested is different than what the authors provided in Fig 4, specifically I was suggesting “correlation between all cytokines simultaneously (i.e. each row and column is a cytokine, with the matching value being the correlation between the two); and hierarchical clustering of the rows and columns (to cluster cytokines with similar correlation patterns)” along with TMB and JAK mutations.

2) Are any cytokine states or cytokine levels associated with response, OR differential response?

Data availability – at least processed data (a maf or vcf of the mutational variant calls, variant allele fractions, etc. used in the analysis) should be made available in supplementary data.

Figure 5 – the authors have now described the source of the data for the interaction figure in the response document, but there is no citation in the text or any description of how the figure was generated (for interpretation (e.g. what is the meaning of the length and thickness of each arc between nodes; how were nodes chosen?) and reproducibility)

Re: interferon gamma low and JAK mutations being associated with improved survival under immunotherapy treatment, the authors may wish to review/cite recent papers by the Minn lab (10.1016/j.cell.2019.07.019 and <https://doi.org/10.1038/s43018-022-00490-y>) which show that chronic interferon gamma signalling in tumor cells leads to downregulation of anti-tumor adaptive and innate signalling circuits.

Minor

There are two Supplementary Table 2s

There is no legend for Table 2

Reviewer #4 (Remarks to the Author): new referee with expertise in biostatistics, clinical trials

Here, Ascierto and colleagues present 4-year follow-up from the SECOMBIT trial, a study of sequential immunotherapy and targeted therapy for metastatic BRAF V600-mutated melanoma. This trial was essential in determining the optimal treatment approach (BRAf/MEK inhibition or CTLA-4/PD-1 therapy first) for this patient population, and the third “sandwich” arm was a novel addition as a potential treatment approach compared to the DREAMseq trial. While the survival follow-up information is important to the field, there are some design flaws regarding the hypothesis generating biomarker analysis conducted. The findings regarding JAK mutations and serum interferon gamma could be of great utility to the field and are novel in the literature, however, the statistical rigor of these findings is in question. The methodology used to deduce the results needs to be elaborated upon and improved, and additional details are needed for this work to be reproducible. See below for specifics:

1) The term “trend” is used throughout the paper to refer to patient survival relationships with serum interferon gamma (IFN γ) and JAK mutation status, however, the p-values for the tests of the JAK mutated vs. not mutated and the IFN γ are not close to statistical significance. Consider removing this term or selecting a different phrase.

2) Why were only JAK mutation status and IFN γ highlighted throughout this paper? Is there literature supporting investigation of these specific elements within this disease? Information seems to have been collected on many mutations/genomic pathways and cytokines, however, only the relationship of these two with patient survival are reported. Were the other mutations and cytokines investigated? Selectively reporting results where a signal is seen biases the results. If other mutations and cytokines were investigated but had no relation to patient survival, the null results should be discussed as well.

3) Why are p-values not included in Table 1 to compare clinical characteristics across treatment arms? Consider adding this to show balance across the randomized groups

4) Line 173 states that NGS was performed on 28 patients from Arm A, but the manuscript later states that 5 patients had JAK mutated and 24 wild-type in Arm A, which adds to 29. Please clarify.

5) Table S1 is stated to contain JAK mutation status and serum interferon gamma information, however, this is not what is presented in the table. Please either add a table with this information and correctly reference it or remove the references to this table.

6) Why are no p-values presented for Figures 2 and 3 within each treatment arm? Permuted log-rank tests can be used, which is a test typically employed when sample sizes and number of events are small.

7) Why was ROC analysis used to identify a cutpoint rather than using maximally selected rank statistics (the “maxstat” package in R)?

8) I see that a previous reviewer request that IFN γ be evaluated continuously to evaluate its relationship to survival in a Cox model, however, since Figure 3 presents the dichotomized values, I think it would be better to test the cut-off value (or include IFN γ evaluated both ways).

9) How can the JAK mutation status, TMB and LDH going in the opposite direction for Arm C compared to Arms A and B be interpreted?

10) The correlation between INF γ and VEGFC in Arm A is reported as positive with a value of 0.603, however, lines 238-239 report it is negatively correlated. Please rectify.

11) Were p-values in table 2 adjusted for multiple comparisons? A lot of p-values are reported here so this is important to do. Why is Arm B not reported? In the rebuttal the authors also state that the combined results across treatment Arms are reported but I do not see them.

12) Why was IL-8 left out of Table 2?

13) Why does the legend go from -2 to 2 in figure 4? What does this scale represent? What method was used for the co-clustering? Further detail needs to be provided on the methodology used to generate this figure, as well as its interpretation.

14) The statistical analyses section is inadequate. OS and PFs need to be defined. What were the start and end times used for these analyses? It needs to be mentioned that Cox proportional hazard models were used to get the hazard ratios. What tests were used in Table S1 and S2-- log rank or Cox proportional hazard p-values? Explain the methodology used for Figure 4 here.

15) Why for specific gene pathways was a more accurate analysis with less stringent parameters performed? How were these pathways chosen?

16) Why is what is labeled as Table S2 in the main document and not the supplement? There is a Table S2 already in the supplement so this needs to be re-labelled.

Reviewer's Comments:

Reviewer #2 (Remarks to the Author)

Ascierto et al have now improved their manuscript significantly in response to the concerns raised in the original review and provide important new data on both survival updates of the trial and exploratory biomarkers that differentiate response. However, I have a few remaining questions and recommendations, focused both in statistics, presentation, and in the exploratory biomarker analysis.

I clearly appreciate the bind the authors are in – the study was not designed or powered to compare the different arms, or further to do subgroup analyses, and thus analysis of differences between arms and in subgroups is exploratory as is now clear in the text. However, the results as presented are quite challenging for the reader to parse through, and statements in the discussion, e.g. “Low serum levels of interferon at baseline as well as mutations in JAK were associated with improved 4-year survival in the sandwich and immunotherapy-first treatment arms” require some type of (statistical) evidence to support the statement. I have two suggestions

- Inference of effect size and confidence intervals without p-values: a Cox PH analysis to generate estimated hazard ratios (as done in their JCO paper reporting earlier results) and CI would be helpful.

Answer: A Forrest plot was produced, and is shown in Figure 2

- A forest plot would help summarize results in overall and (biomarker-driven) subgroups, e.g. Figure 3 from [https://doi.org/10.1016/S0140-6736\(22\)00562-1](https://doi.org/10.1016/S0140-6736(22)00562-1)

Answer: a forest plot has been added (Figure 2)

Other specific questions/suggestions below:

- My preference would be to move TPFS survival curves from supplementary figures to main figures (e.g. Fig 1a and b)

Answer: Figure 1 now presents TPFS and OS in the overall population

- If you provide p-values/comparison, a clear description of the comparison and test performed should be provided (e.g. “...(a log-rank (?) test of difference in survival between TMB-H vs TMB-L tumors across all arms)...”, lines 186-187; similarly in lines 195-196).

Answer: The SECOMBIT trial was not designed as a comparative trial and no P value calculation was planned. We reported all outcomes as a punctual estimate with the associated 95% confidence intervals to facilitate interpretation.

- Line 217 –,”The ROC analysis was based on patient status (alive/dead).” Alive or dead at what time point (4 years?)? How were censored patients considered?

Answer: We thank the reviewer for this question. Censored patients were considered alive. We repeated the cut-off calculation maximizing the log-rank test between the two curves and we obtained the same value.

- I appreciate that the authors have now provided the PolyPhen-2 score and an explanation of the interpretation, as well as JAK1/2/3 mutations from Arm B in Table S2. What is still not clear is whether the authors classified > a score of 0.15 (“possibly damaging”) or > a score of 0.85 (“confidently damaging”) as “deleterious mutations”, or some other threshold. What should also be cited as a limitation is that the NGS is tumor-only targeted panel sequencing, which we know results in false positive calls (e.g. germline SNPs despite large germline filtering datasets without a matched normal to filter against) and systematically higher TMB (e.g. DOI: 10.1200/PO.19.00171).

Answer: The concept is that the closer it is to the threshold 1 the greater the possibility that the mutation is deleterious, in Table S2 there are all the scores of the polyphen score, in which it can be observed that most of them are equal to 1 or very close ...it can therefore be deduced that in most of them a deleterious mutation is very probable.

The patient-level biomarker and outcome information is greatly appreciated and will be very helpful.

- Is there interferon gamma information as well? (specific levels would be best)

Answer: cut-off for IFN-gamma is 0.58 (line 244)

- A dictionary would be helpful (e.g. what is the column “PD” vs “Progr total”).

Answer: thank you for this observation. This table was removed. We checked that full names for all acronyms are present in the manuscript

- Notes about missing values/excluded patients would be helpful

Answer: Tumor stage was not known for one patient in arm B and one in arm C (line 119).

- Data to be able to reproduce Fig 4 would be terrific.

Answer: This approach does not provide directly the data used, but all the paper used to draw the network are available on the web site

- Is there age, gender, stage information that would be available?

Answer: see Table 1. We are analyzing these variables and their correlation with clinical features and outcomes, with a longer follow-up; these analyses will be presented in a future manuscript.

I continue to be confused about the inference of Table 2, which shows differential correlation between ifn-gamma and other cytokines between arms. If there are indeed differences between cytokine correlations between randomized arms before treatment, how do the authors explain/interpret this? Given that these are randomized arms and before treatment, my suggestion would be to examine the cytokine co-correlations across the whole cohort rather than cite differences between arms. This may result in more stable/clear clusters of cytokines that may provide insight into the relevant immune states (e.g. both CXCL10 and IL10 have a positive correlation with IFN-gamma in both arms cited, though only one is statistically significant; IL1b_IL1F2 is not statistically significantly associated with IFN-gamma but is positively associated in both arms and may be associated when considering the entire cohort).

Answer: We thank the reviewer for this observation and we produced Table 2 for the whole cohort

In my mind, there are two distinct questions:

1) Are there specific immune states reflected by cytokine profiles that cluster (e.g. IFN-G, CXCL10, IL-10 are co-correlated)? (The co-correlational clustered heatmap which I suggested is different than what the authors provided in Fig 4, specifically I was suggesting “correlation between all

cytokines simultaneously (i.e. each row and column is a cytokine, with the matching value being the correlation between the two); and hierarchical clustering of the rows and columns (to cluster cytokines with similar correlation patterns)” along with TMB and JAK mutations.

Answer: clusters in the three arms are now presented in Figure 5 (previously 4) B,C,D

2) Are any cytokine states or cytokine levels associated with response, OR differential response?

Answer: We found that MMP-9 and IFN- α were significantly more expressed in patients with SD or PD than in those with PR or CR. We added this in the manuscript and provide Figure S5

Data availability – at least processed data (a maf or vcf of the mutational variant calls, variant allele fractions, etc. used in the analysis) should be made available in supplementary data.

Answer: Variant allele frequency (VAF) file was uploaded on following

URL: <https://doi.org/10.5281/zenodo.8386539>; We estimated the proportion of sequence reads that matched a particular DNA variant by dividing this count by the overall number of reads at the relevant genomic locus in order to obtain the Variant Allele Fraction (VAF) for each somatic variant recovered from VCF files. The following formula was used by us:

VAF = (N_ref + N_var)/N_var; By counting reads that supported both the reference allele (N_ref) and the variant alleles (N_var) at the pertinent genomic region, we were able to determine the results.

Figure 5 – the authors have now described the source of the data for the interaction figure in the response document, but there is no citation in the text or any description of how the figure was generated (for interpretation (e.g. what is the meaning of the length and thickness of each arc between nodes; how were nodes chosen?) and reproducibility)

Answer: the data present in the STRING database are generated by the comparison of numerous papers concerning the proteins in question. Line thickness indicates the strength of data support

Re: interferon gamma low and JAK mutations being associated with improved survival under immunotherapy treatment, the authors may wish to review/cite recent papers by the Minn lab (10.1016/j.cell.2019.07.019 and <https://doi.org/10.1038/s43018-022-00490-y>) which show that chronic interferon gamma signalling in tumor cells leads to downregulation of anti-tumor adaptive and innate signalling circuits.

Answer: the suggested reference is now cited in the Discussion (line 286).

Minor

There are two Supplementary Table 2s

Answer: They are re-numbered. Supplementary figures are also re-numbered (The first S2 is now included into Figure 1)

There is no legend for Table 2

Answer: Table 2 has been changed and the legend is now present

Reviewer #4 (Remarks to the Author): new referee with expertise in biostatistics, clinical trials

Here, Ascierio and colleagues present 4-year follow-up from the SECOMBIT trial, a study of sequential immunotherapy and targeted therapy for metastatic BRAF V600-mutated melanoma.

This trial was essential in determining the optimal treatment approach (BRAF/MEK inhibition or CTLA-4/PD-1 therapy first) for this patient population, and the third “sandwich” arm was a novel addition as a potential treatment approach compared to the DREAMseq trial. While the survival follow-up information is important to the field, there are some design flaws regarding the hypothesis generating biomarker analysis conducted. The findings regarding JAK mutations and serum interferon gamma could be of great utility to the field and are novel in the literature, however, the statistical rigor of these findings is in question. The methodology used to deduce the results needs to be elaborated upon and improved, and additional details are needed for this work to be reproducible. See below for specifics:

1) The term “trend” is used throughout the paper to refer to patient survival relationships with serum interferon gamma (IFN γ) and JAK mutation status, however, the p-values for the tests of the JAK mutated vs. not mutated and the IFN γ are not close to statistical significance. Consider removing this term or selecting a different phrase.

Answer: $p < 0.10$ was considered as “suggestive” (lines 207 and 233)

2) Why were only JAK mutation status and IFN γ highlighted throughout this paper? Is there literature supporting investigation of these specific elements within this disease? Information seems to have been collected on many mutations/genomic pathways and cytokines, however, only the relationship of these two with patient survival are reported. Were the other mutations and cytokines investigated? Selectively reporting results where a signal is seen biases the results. If other mutations and cytokines were investigated but had no relation to patient survival, the null results should be discussed as well.

Answer: the analyzes were carried out on all the molecules present in the panels, only the statistically significant molecules were reported in the text. This is now reported in the Discussion (line 276)

3) Why are p-values not included in Table 1 to compare clinical characteristics across treatment arms? Consider adding this to show balance across the randomized groups

Answer: Calculation of p-values for baseline characteristics in randomized clinical trials is controversial, but is mainly discouraged. Many high-impact journals, such as NEJM, consider calculation of p values for baseline characteristics of trial arms as inappropriate and also CONSORT guidelines suggest to avoid it. Randomization, if correct, ensures that differences between arms are due to chance by definition, So, testing is useless. (Stroke. 2014 December ; 45(12): e244–e246, European Journal of Preventive Cardiology, 2011, 19(2) 231–232, Altman DG, 1990, The Lancet, v.335, p.149-150, CONSORT BMJ 2010;340:c869).

4) Line 173 states that NGS was performed on 28 patients from Arm A, but the manuscript later states that 5 patients had JAK mutated and 24 wild-type in Arm A, which adds to 29. Please clarify.

Answer: 29 is the correct number. It is corrected in the manuscript (line 181)

5) Table S1 is stated to contain JAK mutation status and serum interferon gamma information, however, this is not what is presented in the table. Please either add a table with this information and correctly reference it or remove the references to this table.

Answer: Table S1 was changed. It now presents significance of differences in OS and TPFs

6) Why are no p-values presented for Figures 2 and 3 within each treatment arm? Permuted log-

rank tests can be used, which is a test typically employed when sample sizes and number of events are small.

Answer: P values were not calculate because the study was not powered to assess differences of treatment arms. We reported 95% confidence intervals to describe differences along time.

7) Why was ROC analysis used to identify a cutpoint rather than using maximally selected rank statistics (the “maxstat” package in R)?

Answer: Thank you for the suggestion. We repeated the analysis using the maxstat package and the result was exactly the same (i.e. 0.58). This is now mentioned in the manuscript (line 233).

8) I see that a previous reviewer request that IFN γ be evaluated continuously to evaluate its relationship to survival in a Cox model, however, since Figure 3 presents the dichotomized values, I think it would be better to test the cut-off value (or include IFN γ evaluated both ways).

Answer: The HR of IFN gamma evaluated continuously was 1.11 (0.95-1.30), and HR for the dichotomized evaluation was 1.93 (0.99-3.76) (line 246).

9) How can the JAK mutation status, TMB and LDH going in the opposite direction for Arm C compared to Arms A and B be interpreted?

Answer: Arm C is characterized by an initial administration of targeted therapy followed by immunotherapy. We hypothesize that this short initial administration of targeted therapy could modify some biological mechanisms underlying a resistance to immunotherapy. (lines 338-340).

10) The correlation between IFN γ and VEGFC in Arm A is reported as positive with a value of 0.603, however, lines 238-239 report it is negatively correlated. Please rectify.

Answer: Table 2 was changed as suggested by another reviewer, and now shows correlations in the overall population. The text has been corrected (lines 249-52).

11) Were p-values in table 2 adjusted for multiple comparisons? A lot of p-values are reported here so this is important to do. Why is Arm B not reported? In the rebuttal the authors also state that the combined results across treatment Arms are reported but I do not see them.

Answer: Table 2 now shows correlations in the overall population and was corrected for multiple testing. We are available to move this table to supplementary material.

12) Why was IL-8 left out of Table 2?

Answer: IL-8 was mentioned in the Cytokine analysis section of Methods by mistake. It was not present in the analysis.

13) Why does the legend go from -2 to 2 in figure 4? What does this scale represent? What method was used for the co-clustering? Further detail needs to be provided on the methodology used to generate this figure, as well as its interpretation.

Answer: The elements of a heat map are colored according to the value of each variable for each sample. The color scale is adjusted to the choice of normalization method as well as

limited to minimize the impact of potential outliers, the setting for the visible lower and upper scale bounds is two standard deviations. The color scale that is used is shown in the Color Legend. ANOVA was used for comparison of groups.

Note that in order to limit the influence of outliers, the color scale by default does not cover the entire range of data values. The scale limits can however be modified in the Plot Settings dock window

The legend of the figure has been improved.

14) The statistical analyses section is inadequate. OS and PFs need to be defined. What were the start and end times used for these analyses? It needs to be mentioned that Cox proportional hazard models were used to get the hazard ratios. What tests were used in Table S1 and S2-- log rank or Cox proportional hazard p-values? Explain the methodology used for Figure 4 here.

Answer: The Methods section has been improved. Statistical analyses: PFS was calculated as the time between randomization and evidence of relapse or death, whichever occurs first or censored at the time of last evaluation. OS was calculated as the difference between randomization and death, or censored at the time of last follow-up. The Kaplan–Meier method was used to estimate OS and PFS. Statistical analysis was performed using IBM-SPSS version 21.0 or later and R v.4.02 on a Windows 10 operating system.

15) Why for specific gene pathways was a more accurate analysis with less stringent parameters performed? How were these pathways chosen?

Answer: the molecules that correlated significantly with the outcome were initially evaluated, subsequently we evaluated these single molecules in which biological pathway they are involved. (lines 185-186)

16) Why is what is labeled as Table S2 in the main document and not the supplement? There is a Table S2 already in the supplement so this needs to be re-labelled.

Answer: both Tables S1 and S2 are now in the main document

Figure 4 – I have no idea how this figure was generated or what data it is based on.
We appreciate the opportunity to clarify.

Answer: The figure was generated by the STRING database (doi: 10.1093/nar/gky1131)

Cytokine analysis

- It is unclear to me why correlations between ifng and other cytokines at baseline (i.e. before any treatment) should differ between arms, as the arms are randomized. To me, this suggests that any interferon-gamma related program should be analyzed on the basis of combining all patients from all arms.

Answer: In each arm of the trial, patients receive a different treatment sequence (in the SECOMBIT we demonstrate that patients who start the treatment with immunotherapy have better outcome), the analysis of the biomarkers has been evaluated in a sample from each trial arm, according to the treatment (e.g. pz with low/high cytokine XXX could have better outcome if start treatment with immune) for this reason it has been separated for each arm.

- I might suggest a figure that is a heatmap showing correlation between all cytokines simultaneously (i.e. each row and column is a different cytokine, with the matching value being the correlation between the two); and hierarchical clustering of the rows and columns (to cluster cytokines with similar correlation patterns), e.g. Fig 2h of <https://doi.org/10.1038/s41591-019-0654-5>

Answer: The suggested heatmap has been produced and is now presented in Figure 5 (previously 4)

Interaction with prognostic markers of aggressive disease

- What is striking is that OS is very close (non- statistically significant?) at 4 years between Arm A, B, C in the < 3 metastatic site subgroup, despite PFS still being substantially lower in the Combo T group, perhaps suggesting that this is a subgroup with similar outcomes with any of these three strategies, but will require additional validation in independent cohorts (e.g. DreamSeq trial).

Answer: We agree with the reviewer that data must be evaluated in a larger cohort, with longer follow-up, to 5 years. Here we report preliminary results showing trends favoring arms A and C, in terms of OS and even more of TPFS

- OS always lags PFS, so a difference of 5-10% is not surprising. However, there is a very large difference in OS and TPFS specifically in the targeted therapy first (Combo-T) Arm, specifically for < 3 met sites (4 yr TPFS 33% vs OS 55%), and in elevated LDH (4 yr TPFS 18% vs OS 42%) and normal LDH (4yr TPFS 31% vs OS 53%) [lines 148-168, SFig 3,4]. Who are the 20% of patients progressing on targeted -> combo IO therapy who nonetheless have prolonged OS after progression? Does this suggest anything?

Answer: The clinical trial included only the two treatments, targeted therapy and immunotherapy, but the patients who had sustained OS after the trial treatment possibly underwent further subsequent therapies. We are currently gathering this information in order to finalize a new manuscript to answer this question.

Biomarker analysis

- Targeted panel sequencing of 409 genes (tumor-only?) from 25-30 pts from each arm

- Pre-treatment peripheral blood cytokines assessed from 20-30 pts from each arm (overlapping? With targeted panel sequencing)

- I'm concerned about how the cutoff for interferon-gamma high/low was chosen – it is described as based on ROC analysis that maximizes sensitivity and specificity (Youden J index) – what is the metric measured for sensitivity and specificity (i.e. what is “positive” vs “negative” for a patient?)? Specifically I worry that this cutoff is chosen post-hoc to maximize separation between survival curves. A less biased approach would be to split at the median, or split based on data distributions (e.g. after observing a natural bimodal distribution in the data). A cox PH analysis would further examine the effect of interferon-gamma across the spectrum of values in an unbiased manner.

Answer: We acknowledge that separation by median is commonly used but not always the best method. In this study we preferred to dichotomize the data selecting the best cut-point using ROC curve, which from a statistical point of view is a widely recognized method. We agree with the reviewer that it could be biased but, due to the experimental nature of this analysis and to the small sample size, we aimed to find the best cut-off to be tested in future independent series. Finding the most appropriate cut-off for a biomarker is always a long process that needs validation on external data and, in our case, should be seen as a starting value to be confirmed.

- How polyphen-2 scores are interpreted as damaging JAK mutations is unclear (e.g. what threshold was used to define deleterious JAK mutations? I assume nonsense mutations are defined as deleterious? Was a single mutation sufficient or were biallelic alterations (e.g. LOH + SNV or two SNVs) considered?
- The per-arm JAK mutational analysis is underpowered, so conclusions are hard to draw; more formally, the presence of a deleterious JAK mutation could be tested for association with survival with a COX PH analysis across the entire cohort and tested for interaction with treatment type.

Answer: The PolyPhen-2 score predicts the possible impact of an amino acid substitution on the structure and function of a human protein. This score represents the probability that a substitution is damaging. Ion Reporter™ Software reports the pph2-prob PolyPhen-2 score.

The PolyPhen-2 score ranges from 0.0 (tolerated) to 1.0 (deleterious). Variants with scores of 0.0 are predicted to be benign. Values closer to 1.0 are more confidently predicted to be deleterious. The score can be interpreted as follows:

- **0.0 to 0.15 -- Variants with scores in this range are predicted to be benign.**
- **0.15 to 1.0 -- Variants with scores in this range are possibly damaging.**
- **0.85 to 1.0 -- Variants with scores in this range are more confidently predicted to be damaging.**

REVIEWERS' COMMENTS

Reviewer #2 (Remarks to the Author):

Ascierto et al have significantly improved their manuscript with the last resubmission with new/improved figures that significantly clarify their findings and greater clarification on their statistical methods. Overall this is an important trial and data, and intriguing findings for exploratory biomarker analysis. I have some remaining questions/comments/suggestions to clarify their message and findings.

Lines 130-134: "During treatment across the arms, there were 13 deaths in arm A... 11 deaths in Arm B... and 4 deaths in Arm C..." – on reread, I'm confused by this statement. Clearly (e.g. from the overall survival graph, Fig 1B) there were many more deaths than reported here (e.g. more than 4 deaths in Arm C). Would you clarify this statement?

Figure 2 is quite informative but needs clearer labeling

- can you put in a vertical line at HR = 1?
- Can you label the direction for the associated categories in the figure (e.g. TMB-high, IFNg-high) and cutoffs in the legend? (e.g. > 10mut/MB, >= 0.58)
- Can you add LDH-high, met sites high to the Forest plot as I think it highlights the interesting differences in Arm C
- The legend needs to be improved/expanded. E.g. Forest plot, HR of (overall survival, TPFs).

Table S1 – I'm confused, there are two A vs B p-values, or perhaps A vs ABC p-values under both Overall Survival and TPFs. Can you clarify in the legend or in the table what these represent?

I'm not entirely sure what or how the tests of interaction were carried out (Lines 239-240, 211-212, 201-202), or what it is testing. Is this a formal test of a differential effect of a biomarker with different treatments? (e.g. testing whether high LDH is differentially associated with survival in Arm C vs the other arms; $\text{Surv} \sim \text{Arm C} + \text{Arm B} + \text{high_LDH} + \text{Arm C} * \text{high_LDH} + \text{Arm B} * \text{high_LDH}$).

For Table 2 (cytokines associated with IFN γ levels) I would provide both nominal and FDR corrected p-values, and also order them (e.g. from highest correlation to least)

Is there no longer patient level data being provided? E.g. TPFS, OS, LDH, interferon-gamma, number of met sites, cytokine levels, response to therapy; needed for reproduction of all the figures (and also for future comparative research and transparency).

Please add a data availability statement with the location of the data.

New Fig 5

- I like (A) which shows all the data for cytokines and clinical features together; it shows for example a cluster of patients with high IL1b_IL1F2, IL6, PDGF, HB (hemoglobin), and what looks like more met sites and high LDH -- am curious if these folks were more or less responsive to therapy (probably insufficient numbers to actually tell).
- There are 4 tumors at about 1/3 of the way across where it looks like ALL cytokine values are exactly at the mean (value of 0) – is this correct? Or are there no cytokine values for these tumors (NA)?
- IL10 values look strangely non-varying (black across all tumors except one) – are there any issues with IL10 values?
- Is “IFN” interferon-gamma? Or one of the other interferon, or a combined metric?
- (B), (C), (D) also appear to be in the Supplementary Figures (S6? There’s no legend for it).

Discussion

– might be worth referring back to the pleiotropic effects of IFN-G referenced in the summary/intro in recent publications (references 9-14) in discussing why low IFN-G and JAK1/2 mutations are associated with better immunotherapy response (Lines 328-333).

Please cite the STRING database in the text as well as how Figure 6 was generated for reproducibility.

Reviewer #4 (Remarks to the Author):

Thank you to the authors for adequately addressing my concerns regarding the manuscript.

I have no further comments.

REVIEWERS' COMMENTS

Reviewer #2 (Remarks to the Author):

Ascierto et al have significantly improved their manuscript with the last resubmission with new/improved figures that significantly clarify their findings and greater clarification on their statistical methods. Overall this is an important trial and data, and intriguing findings for exploratory biomarker analysis. I have some remaining questions/comments/suggestions to clarify their message and findings.

Lines 130-134: "During treatment across the arms, there were 13 deaths in arm A... 11 deaths in Arm B... and 4 deaths in Arm C..." – on reread, I'm confused by this statement. Clearly (e.g. from the overall survival graph, Fig 1B) there were many more deaths than reported here (e.g. more than 4 deaths in Arm C). Would you clarify this statement?

Answer: this sentence reports deaths during treatment; the additional deaths occurred after treatment discontinuation (CONSORT diagram and Supplementary Figure 1).

Figure 2 is quite informative but needs clearer labeling

- can you put in a vertical line at HR = 1? - Can you label the direction for the associated categories in the figure (e.g. TMB-high, IFNg-high) and cutoffs in the legend? (e.g. > 10mut/MB, >= 0.58)

Answer: an improved Figure 2 is provided

- Can you add LDH-high, met sites high to the Forest plot as I think it highlights the interesting differences in Arm C

Answer: LDH-high and met site were not added because this figure is related to a small subgroup of patients, while data on LDH and met are those of the whole population

- The legend needs to be improved/expanded. E.g. Forest plot, HR of (overall survival, TPFS).

Answer: the legend has been improved "Figure 2. Forest plot representing HR for Overall Survival according to Jak mutations, IFN gamma expression and TMB in the three arms of SECOMBIT."

Table S1 – I'm confused, there are two A vs B p-values, or perhaps A vs ABC p-values under both Overall Survival and TPFS. Can you clarify in the legend or in the table what these represent?

Answer: the table has been corrected; column partition was mistaken

I'm not entirely sure what or how the tests of interaction were carried out (Lines 239-240, 211-212, 201-202), or what it is testing. Is this a formal test of a differential effect of a biomarker with different treatments? (e.g. testing whether high LDH is differentially associated with survival in Arm C vs the other arms; $\text{Surv} \sim \text{Arm C} + \text{Arm B} + \text{high_LDH} + \text{Arm C} * \text{high_LDH} + \text{Arm B} * \text{high_LDH}$).

Answer: This analysis tested for differential effect of biomarker across arms (i.e. high vs low IFN-gamma in each arm...). We want to remark that SECOMBIT was not powered for comparison in general and, more specifically, only a subgroup of patients were considered for biomarkers so sample size is insufficient to accurately test any interaction.

For Table 2 (cytokines associated with IFNgamma levels) I would provide both nominal and FDR corrected p-values, and also order them (e.g. from highest correlation to least)

Answer: Table 2 has been changed as suggested.

Is there no longer patient level data being provided? E.g. TPFS, OS, LDH, interferon-gamma, number of met sites, cytokine levels, response to therapy; needed for reproduction of all the figures (and also for future comparative research and transparency).

Answer: these data are shared, in the file "SECOMBIT "4yr OS_raw_data_v2" in the zenodo repository zenodo available at URL: <https://doi.org/10.5281/zenodo.8386539>

Please add a data availability statement with the location of the data.

Answer: the statement is present in the manuscript, lines 578-584 (<https://doi.org/10.5281/zenodo.8386539>; URL: <https://doi.org/10.5281/zenodo.8386539>)

New Fig 5

- I like (A) which shows all the data for cytokines and clinical features together; it shows for example a cluster of patients with high IL1b_IL1F2, IL6, PDGF, HB (hemoglobin), and what looks like more met sites and high LDH -- am curious if these folks were more or less responsive to therapy (probably insufficient numbers to actually tell).

Answer: yes, the number of patient was too low for this question

- There are 4 tumors at about 1/3 of the way across where it looks like ALL cytokine values are exactly at the mean (value of 0) – is this correct? Or are there no cytokine values for these tumors (NA)?

Answer: all values are about 0; there are also unavailable cytokines

- IL10 values look strangely non-varying (black across all tumors except one) – are there any issues with IL10 values?

Answer: No, the detected IL-10 values are reported.

- Is "IFN" interferon-gamma? Or one of the other interferon, or a combined metric?

Answer: Yes, it is IFN γ , and the figures have been corrected

- (B), (C), (D) also appear to be in the Supplementary Figures (S6? There's no legend for it).

Answer: the supplementary file has been changed. Five supplementary figures are present.

Discussion

– might be worth referring back to th pleiotropic effects of IFN-G referenced in the summary/intro in recent publications (references 9-14) in discussing why low IFN-G and JAK1/2 mutations are associated with better immunotherapy response (Lines 328-333).

Answer: this comment has been added (line 318)

Please cite the STRING database in the text as well as how Figure 6 was generated for reproducibility.

Answer: "STRING is a database of known and predicted protein-protein interactions. The interactions include direct (physical) and indirect (functional) associations; they stem from computational prediction, from knowledge transfer between organisms, and from interactions aggregated from other (primary) databases." This has been added to the figure legend with references.